# Efficient tool segmentation for endoscopic videos in the wild

**Clara Tomasini**[1]                                         CTOMASINI@UNIZAR.ES
**Iñigo Alonso**[1]                                              INIGO@UNIZAR.ES
**Luis Riazuelo**[1]                                          RIAZUELO@UNIZAR.ES
**Ana C. Murillo**[1]                                              ACM@UNIZAR.ES
[1] *Universidad de Zaragoza, Zaragoza, Spain*

## Abstract

In recent years, deep learning methods have become the most effective approach for tool segmentation in endoscopic images, achieving the state of the art on the available public benchmarks. However, these methods present some challenges that hinder their direct deployment in real world scenarios. This work explores how to solve two of the most common challenges: real-time and memory restrictions and false positives in frames with no tools. To cope with the first case, we show how to adapt an efficient general purpose semantic segmentation model. Then, we study how to cope with the common issue of only training on images with at least one tool. Then, when images of endoscopic procedures without tools are processed, there are a lot of false positives. To solve this, we propose to add an extra classification head that performs binary frame classification, to identify frames with no tools present. Finally, we present a thorough comparison of this approach with current state of the art on different benchmarks, including real medical practice recordings, demonstrating similar accuracy with much lower computational requirements.

**Keywords:** tool segmentation, endoscopy, real-time, recordings in the wild

## 1. Introduction

In recent years, machine learning techniques, and more specifically convolutional neural networks (CNN), have been widely used for analysis of medical images, including endoscopic images. The main goal in using these techniques is to aid and improve diagnosis through detection and classification of specific structures and characteristics related to certain diseases (Choi et al., 2020). Some applications of such techniques include automatic detection and classification of gastric cancers (Hirasawa et al., 2018) and polyps (Zhang et al., 2016) using gastroscopy or colonoscopy, as well as classification of different anatomical locations in esophagogastroduodenoscopy images (Takiyama et al., 2018). As a result of medical specialists using these tools as an aid to complement their skills and knowledge, some diseases could be detected earlier and better treated, improving the patient's condition.

This work is focused on the particular problem of tool segmentation in endoscopic images, commonly solved by machine learning techniques and CNNs. More specifically, it is a binary semantic segmentation problem that, as described in (Münzer et al., 2018), is a key pre-processing step for automatic scene understanding, to facilitate downstream tasks such as monitoring, augmented reality or 3D reconstruction. It has got plenty of attention in the field, for example, it was the main focus of the 2017 Robotic Instrument Segmentation Challenge (EndoVis17) (Allan et al., 2019). Figure 1 shows three examples of the tool segmentation results on different datasets using the MiniNet architecture (Alonso et al., 2020) applied in our work.

The problem of tool segmentation in endoscopic sequences presents several challenges. We focus on two significant ones in this work. The first challenge is processing endoscopic images *in the wild*, i.e. complete endoscopic videos acquired during real medical practice. Two subtasks appear in trying to solve this challenge. Firstly, in full endoscopy sequences, many frames don't actually contain tools. Therefore, when the segmentation model is applied to these frames, it introduces *false positives*. These *false positives* have to be removed to allow for any further processing using the segmented images, since they can cause wrong results which could be easily avoided. The second subtask comes from the fact that endoscopic images present very specific characteristics such as lighting and colors, that have a strong dependency on the endoscope model, protocols followed by the specialists, etc. Frequently, segmentation models developed for a certain medical dataset cannot be used directly on another dataset, and need to be adapted to the specific target data distribution using *domain adaptation*. Besides, this particular task may have specific requirements, and we find several segmentation models developed specifically for medical image segmentation, such as UNet (Ronneberger et al., 2015), MF-TAPNet (Jin et al., 2019), DMNet (Wang et al., 2021). General purpose models can also be successfully adapted to endoscopic images, as shown in Endovis17 challenge winning solution (Shvets et al., 2018) with TernausNet (Iglovikov and Shvets, 2018) or LinkNet (Chaurasia and Culurciello, 2017), and as we do in this work .

Another main challenge consists in enabling the use of the *segmentation model in real-time*. Running segmentation models in real-time is essential when the segmentation is part of a downstream task with computational restrictions, such as 3D reconstruction and mapping tasks targeted, for instance, in the EndoMapper project (Endomapper, 2021). Efficiency in segmentation models is an active area of research, with approaches such as MiniNet (Alonso et al., 2020) and DMNet (Wang et al., 2021). DMNet has been developed for medical tool segmentation and gives very accurate segmentations when applied to endoscopic images, but the number of parameters and inference time are still far from the real-time use case. Our recently developed architecture, MiniNet, presents significantly lower computational requirements and promising properties for quick domain adaptation, but so far it has been evaluated on urban scenarios.

This paper targets more efficient tool segmentation in complete endoscopic videos acquired during real medical practice, i.e., *in the wild*. Our contributions are the following:

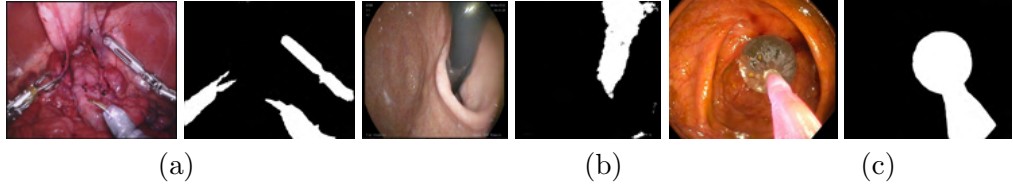

(a)                   (b)                   (c)

Figure 1: Examples of tool segmentation results with MiniNet architecture for different benchmarks of endoscopic images: (a) EndoVis2017 dataset (b) Kvasir-Instrument dataset (c) HCULB-tools dataset

- A thorough study on the **performance of the MiniNet model in endoscopic datasets**. We show high accuracy (similar to current state of the art) and significantly lower computational requirements than models currently used in this field.

- A **novel module**, integrated within the MiniNet architecture, to achieve a quick binary frame classification of *tools* vs *no-tools* frame. Our modification to MiniNet consists of an **additional classification head** that serves as quick pre-filtering of images that contain no tools. We show how this helps us avoiding plenty of false positives when running on complete sequences of real medical procedures.

## 2. Methodology

This section describes the different segmentation models used in this work, with a more detailed summary of the MiniNet architecture and the proposed novel binary-module just mentioned in previous section.

### 2.1. Pre-existing models used for medical segmentation

In this work, we evaluate five models that were originally designed to be used for segmentation either in medical images or other types of images. The more general purpose models were then adapted for and evaluated on medical benchmarks. Within these models, three of them are UNet-based models, and the other two use temporal as well as semantic information to compute the segmentation masks.

**UNet-based models.** These three models were compared in the winning paper (Shvets et al., 2018) of EndoVis17 Challenge (Allan et al., 2019).

- **UNet (Ronneberger et al., 2015):** UNet is a fully convolutional network specifically designed for semantic segmentation of biomedical images. It is composed of a contracting path and an expanding path. Both paths are formed by a succession of convolutional and pooling (contracting path) or upsampling (expanding path) layers.

- **TernausNet (Iglovikov and Shvets, 2018):** It follows the UNet architecture but uses VGG-16 (Simonyan and Zisserman, 2014) pre-trained on ImageNet as the encoder. It was originally developed as a more general purpose segmentation model and later evaluated on medical images in (Shvets et al., 2018), with very good results for tool segmentation.

- **LinkNet (Chaurasia and Culurciello, 2017):** Similarly to TernausNet, it uses ResNet-34 (He et al., 2016) pre-trained on ImageNet as the encoder.

**Semantic + temporal models.** Two more recent models, developed specifically for tool segmentation in endoscopy, are MF-TAPNet (Jin et al., 2019) and DMNet (Wang et al., 2021). These models include temporal as well as semantic information extracted from the input images to compute the segmentation masks.

- **MF-TAPNet (Jin et al., 2019):** MF-TAPNet uses motion flow between consecutive frames calculated with Unflow (Meister et al., 2018) model to compute maps

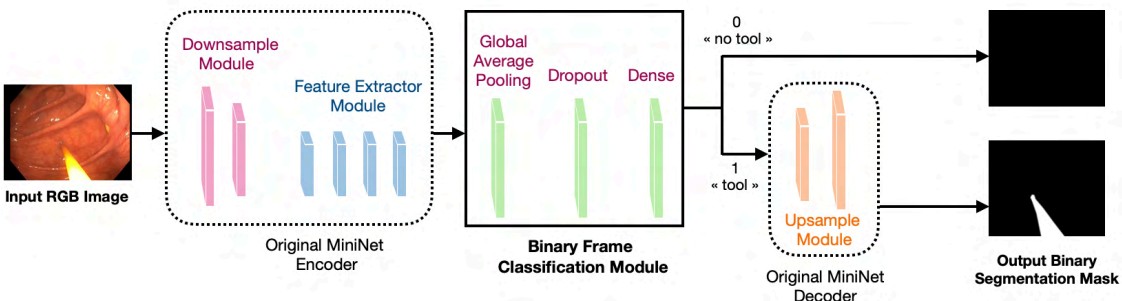

Figure 2: Proposed pipeline: MiniNet + binary frame classifier for pre-filtering

representing displacement between two consecutive frames. These maps can be combined with the computed segmentation mask to obtain current frame segmentation.

- **DMNet (Wang et al., 2021):** DMNet follows the same idea but with a Dual-Memory system to include temporal information from several previous frames. This reduces the model parameters and inference time and facilitates its use in real-time.

### 2.2. Our approach: MiniNet + Binary frame classification head

Our approach [1] to real-time semantic tool segmentation is based on our recently proposed architecture, MiniNet (Alonso et al., 2020). It is a segmentation model designed for general purpose semantic segmentation, and has not been evaluated on medical images. Its architecture follows an encoder-decoder structure formed by four convolutional blocks (downsample, feature extractor, refinement and upsample). It focuses on efficiency, facilitating its use in real-time applications. Higher efficiency is achieved using multi-dilation depthwise separable convolutional layers in the encoder, reducing the number of parameters and amount of memory needed by the model. It achieves similar or better results than state-of-the-art efficient architectures on benchmark non-medical datasets. The low computational requirements lead to lower energy consumption, CO2 equivalent emissions and carbon footprint caused by the model, making it a relatively environmental friendly model.

**Binary frame classification module (*pre-filter*).** When applied in real medical practice, the segmentation model needs to process the whole endoscopic recording, including plenty of frames where there is no tool. As it could be expected, this leads to a significant amount of false positives with any of the existing models evaluated. These false positives could be removed or prevented with other strategies, but we have explored them and they do not provide as good results as the proposed strategy. If we train the models including a lot of frames with no tools, the resulting model does not reach the same level of accuracy. If we attempt to remove the false positives by post-processing the segmentations (based on shape or location of the segmented regions), it would be feasible, but not efficient at all, since all frames from the sequence would have to be processed completely by the segmentation model, and then by the post-processing module.

---

1. Code at https://github.com/endomapperUZ/toolSegmentation

We propose to include a simple additional step integrated on the segmentation architecture of MiniNet. It consists of an additional binary frame classification head to determine if tools are present or not in a given frame. Note this is done before running the whole segmentation. We use the encoder blocks of MiniNet as feature extractor, and train separately a simple binary classification layer (*tools vs no-tools*) for this new step. Figure 2 sums up the proposed pipeline composed of MiniNet and our binary frame classification module. The proposed binary classifier consists of a Global Average Pooling layer, a Dropout layer with rate 0.2 and a Dense layer. For training we use the Adam optimizer and Binary Cross-Entropy $CE(\boldsymbol{y}, \hat{\boldsymbol{y}}) = -\sum_{i=1}^{N_c} y_i \log(\hat{y}_i)$, where $N_c$ is the number of classes, $y$ is the real label and $\hat{y}$ the predicted label. The new layers of the binary frame classifier are trained on the dataset needed, while the weights of the Feature Extractor block are loaded from MiniNet pre-trained on said dataset and then frozen.

## 3. Experimental set up

**Datasets.** Our evaluations use the following datasets:

***EndoVis2017 dataset*** (Allan et al., 2019): Publicly available dataset (developed for the 2017 Robotic Instrument Segmentation Challenge): eight real endoscopic videos (225-frames each) with ground truth segmentation of the surgical tools that appear in each frame.

***Kvasir-Instrument dataset*** (Jha et al., 2021): Publicly available dataset composed of 590 frames obtained during real gastrointestinal endoscopic procedures as well as corresponding tool segmentation annotations.

***HCULB dataset*** (Endomapper, 2021): Dataset recorded during real medical practice by the EndoMapper project, currently private[2]. It consists of a set of videos of full gastroscopy and colonoscopy procedures captured during real medical practice. Since these sequences were captured during real medical procedures, the main objective while recording was not to get the best suited sequences for training. Then, the frames can be often blurry or include reflections due to non-optimal movements of the camera or lighting conditions. These aspects make the data more challenging to process. We manually labeled frames where tools were present in fragments from four sequences. Three of the sequences are used as training and one as evaluation. From these four sequences, we obtain two subsets : HCULB-tools and HCULB-full. HCULB-tools only contains frames with tools, whereas HCULB-full contains all frames from HCULB-tools and added frames without tools.

**Metrics.** To account for quality and efficiency of the methods we compute the following:

- **Mean Intersection Over Union (mIoU) (%):** The most commonly used metric for evaluating semantic segmentation models, computed as in (Allan et al., 2019).

- **Binary Accuracy (%):** percentage of frames where the predicted label matches the ground-truth label (used to evaluate our binary frame classification module)

- **Inference Time:** Time to compute the segmentation mask of a one image

2. This data is on the process of being released, and we will contribute the manually acquired tool segmentation labels. A more detailed description of this dataset is available in the supplementary material.

- **Number of Parameters** of each model.

**Implementation details.** Training and fine-tuning of the different models are done using a GPU Tesla V100 SXM2 from a computing cluster. Evaluation of the models is done using a GPU GeForce RTX2080 on a desktop. For the experiments where it is necessary to re-train/fine-tune models, we use the training implementation provided by the respective authors and adapt it to allow fine-tuning on a different dataset. More details on all the training and fine-tuning processes run are available in the supplementary material.

## 4. Results

### 4.1. Comparisons on Tool Segmentation benchmarks

This section compares state of the art tool segmentation models on the three endoscopic datasets described in previous section for binary tool segmentation task.

**Binary tool segmentation.** This task consists of labelling every pixel as tool or not. For this experiment, all models were first trained with EndoVis17 training set, and fine-tuned on the additional datasets (HCULB-tools and Kvasir-instrument) respectively. To fairly compare to existing published results, MiniNet is trained as described in (Shvets et al., 2018), using 4-fold cross-validation on EndoVis17 dataset. Table 1 presents the results obtained on each dataset. We can observe how MiniNet gets similar results to state-of-the-art model MF-TAPNet, but requires a lot less memory (less parameters) and presents significantly faster inference. The lower mIoU of all models on HCULB-tools dataset compared to other datasets confirms the higher complexity of HCULB-tools dataset. Fig. 3 shows a few examples of segmentation results on the HCULB-full test set (using models fine-tuned on HCULB-tools dataset and no pre-filtering step).

### 4.2. Tool Segmentation in the wild

Kvasir-Instrument and EndoVis17 datasets, as well as the training data from HCULB-tools dataset used in the first experiment, only include frames which contain tools. Therefore the

Table 1: **Binary segmentation** results (**mIoU**) for models pre-trained on EndoVis17 and fine-tuned for the rest. N.A.: Not available due to computational resource limitations.

| Models | Datasets | | | Performance | |
|---|---|---|---|---|---|
| | **EndoVis17** | **Kvasir-Inst.** | **HCULB-tools** | **Params(M)$^\$$** | **Time(ms)$^+$** |
| U-Net | 75.44 | 85.78 | 55.63* | 7.85 | 54 |
| TernausNet | 83.60 | N.A. | N.A. | 36.92 | 119 |
| LinkNet | 82.36 | 87.75* | 60.54* | 21.79 | 34 |
| MF-TAPNet | 87.56 | 86.81* | 66.87* | 37.73 | 155 |
| MiniNet | 87.16* | 85.13* | 66.65* | 0.52 | 26 |

*: Results for model retrained in this work as explained. Otherwise, mIoU reported by authors.

$^\$$: *Params*: Memory required by the model (M = millions of parameters).

$^+$: *Time*: Average inference time for 1 image on GPU RTX2080.

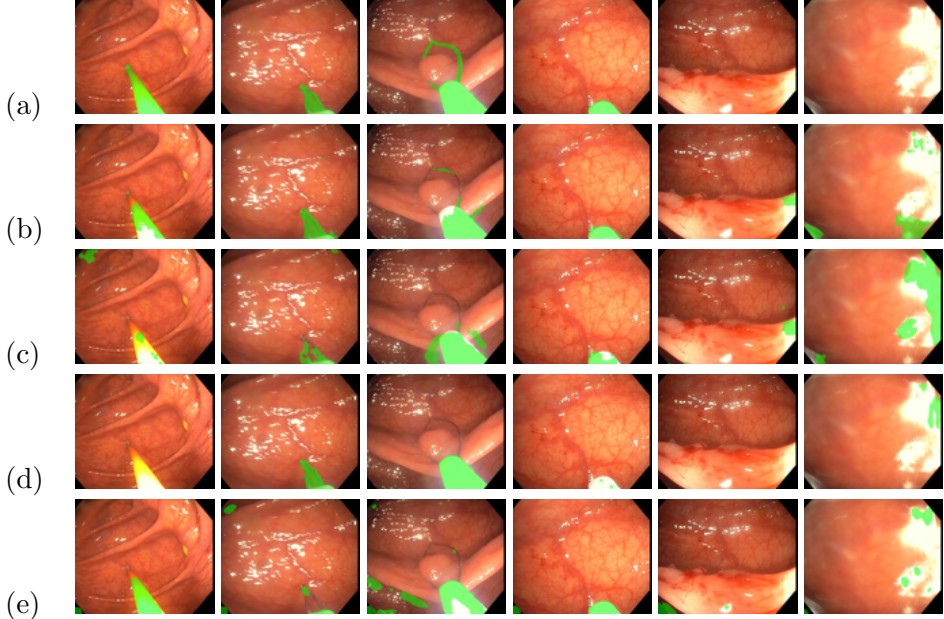

(a)

(b)

(c)

(d)

(e)

Figure 3: **Binary Segmentation** examples from HCULB-full dataset using different approaches fine-tuned on HCULB-tools dataset: (a) Ground-truth manual segmentation (b) MiniNet (c) UNet (d) LinkNet (e) MF-TAPNet

models, including MiniNet, have trained with frames where there is always some tool pixel present.

This produces a lot of noise (*false positives*) in the case of attempting to segment frames where there are no tools, as seen in Fig. 3 in the two right columns. The presence of *false positives* poses a problem when applying the segmentation models to real full colonoscopy sequences, since frames containing tools only represent a small fraction of the whole sequence.

If we re-evaluate the segmentation results on the HCULB-full test set, which contains frames with and without tools, the value of mIoU decreases a lot due to the large number of *false positives*. Table 2, column without pre-filter, shows this decreased accuracy, with respect to previous experiment average values.

Next we discuss how we propose to mitigate this problem.

**Binary frame classification module *pre-filtering* evaluation.**  To limit the amount of memory and time needed by the full pipeline (pre-filtering + segmentation), we develop a binary frame classification module, in particular trained with the HCULB-full data. One of the requirements in this step is to keep it computationally constrained, not to include too many extra computations. This new binary classifier achieves 72.64% accuracy on the test set from HCULB-full dataset and inference from features takes 2ms. We first apply the MiniNet encoder and classifier, then the MiniNet decoder or full MF-TAPNet model.

Table 2 shows the results of this experiment, which demonstrate how the accuracy drop due to the false positives in images with no tools is mitigated with our proposed

Table 2: Binary tool segmentation evaluation (mIoU and inference time) on HCULB-full
sequences using our *pre-filter*.

| Models | mIoU without *pre-filter* (%) | mIoU with *pre-filter* (%) |
|---|---|---|
| MiniNet | 33.21 | 66.09 |
| MF-TAPNet | 18.98 | 51.20 |

pre-filtering step. Only MiniNet and MF-TAPNet are considered as they get the highest mIoU ( 66%) on binary segmentation in previous experiment. The mIoU without applying the pre-filter decreases a lot for both approaches compared to previous experiment (from 66.65% to 33.21% for MiniNet and from 66.87% to 18.98% for MF-TAPNet). Note how after applying the pre-filtering it gets back up to **66.09%** using MiniNet and **51.20** using MF-TAPNet. The models therefore reach mIoU values closer to the previous simpler experiment using only tool frames, and the negative effect of false positives is clearly mitigated. The larger gap between mIoU from not filtered and filtered images in the case of MF-TAPNet is due to the number of *false positives* introduced by the model being higher. Since the classifier only reaches 72.64% accuracy, it can't compensate as well as in the case of MiniNet, which introduces less false positives. Indeed, all segmentation models could benefit from the addition of the binary frame classifier, but since it is integrated with the MiniNet architecture, it presents the most suitable situation with this approach. Note that we are re-using the MiniNet encoder, so if the binary pre-filter is positive, we only need to finish executing the decoder part of Mininet, with the consequent execution time reductions. As previously mentioned, any alternative involving post-processing based on position or shape of the segmentation results to decide whether they look like real tools or just like false positive noise, implies having run the whole segmentation already in every frame. This would therefore increase the processing time compared to our binary frame classifier.

## 5. Conclusion

This work studies the problem of tool segmentation in endoscopic images with an emphasis on efficient models and their performance when running on complete real medical procedure recordings. We have shown how general purpose MiniNet reaches comparable results to state-of-the-art on several benchmarks for tool segmentation while presenting much lower computational requirements (memory and inference time). This aspect makes this model more suitable than current state of the art models for tool segmentation for applications that present time or computational restrictions.

Including tool segmentation solutions in real world applications also involves processing the complete recordings, not separate sets of frames where all of them contain surgical tools in the image. Segmentation models tend to produce significant amounts of false positives in images without any tool. Our work contributes a simple solution to mitigate this issue by incorporating a new classification head to MiniNet that re-uses the encoder part and filters quickly if frames contain tools or not before proceeding with the complete segmentation task.

## Acknowledgments

This project has received funding from the European Union's Horizon 2020 research and innovation programme under grant agreement No 863146 and Aragon Government FSE-T45_20R.

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

## Appendix A. Supplementary Materials

### A.1. Additional training details

Table 3 sums up all the parameters chosen for training of the different models used in this work. The image size used for training is 1056x1280. The MiniNet architecture takes around 8 hours to train, whereas the other networks take twice as much. MiniNet is trained entirely from scratch on the EndoVis17 dataset and then fine-tuned on the other two datasets (Kvasir-Instrument and HCULB). The other networks are available pre-trained on EndoVis17 and we just run the fine-tuning on the same two other datasets.

Table 3: Training parameters for all trained models

| Models | Dataset | Optimizer | Learning Rate | Epochs | Batch Size | Metric |
|---|---|---|---|---|---|---|
| U-Net | HCULB$^t$ | Adam | 1e-3+1e-4* | 10+10* | 2 | mIoU |
| LinkNet | Kvasir-Instrument$^t$ | Adam | 1e-3+1e-4* | 10+10* | 2 | mIoU |
| LinkNet | HCULB$^t$ | Adam | 1e-3+1e-4* | 10+10* | 2 | mIoU |
| MF-TAPNet | Kvasir-Instrument$^t$ | Adam | 3e-5 (initial)$^+$ | 100 max. $^\$$ | 2 | mIoU |
| MF-TAPNet | HCULB$^t$ | Adam | 3e-5 (initial)$^+$ | 100 max. $^\$$ | 2 | mIoU |
| MiniNet | EndoVis17 | Adam | 1e-3 (initial)$^+$ | 20 | 2 | mIoU |
| MiniNet | Kvasir-Instrument$^t$ | Adam | 1e-3 (initial)$^+$ | 20 | 2 | mIoU |
| MiniNet | HCULB$^t$ | Adam | 1e-3 (initial)$^+$ | 20 | 2 | mIoU |
| Binary Classifier | HCULB | Adam | 5e-3 | 20 | 2 | Binary Accuracy |

*: Trained for 10 epochs with $lr = 1e - 3$ and then 10 more epochs with $lr = 1e - 4$

$^\$$: 100 maximum, stops if no change in loss for 20 epochs.

$^+$: Decaying learning rate as per original authors' code

$^t$: Fine-tuning from pre-trained on EndoVis17 dataset.

### A.2. Additional HCULB details

HCULB dataset consists of a large set of colonoscopy and gastroscopy sequences recorded in the context of the project EndoMapper (Endomapper, 2021). These sequences were recorded during real medical practices at HCULB[3]. They have a duration of 20 to 30 minutes and tools appear in some of them. Some of the frames with tools were manually segmented to obtain ground-truth which we use in this work.

The **train set** used in this work consists of 3 videos (ids 39, 181 and 206) from the colonoscopy sequences of the dataset. During training, 80% of the frames from the train set are used for training and 20% are used for validation. The **test set** consist of 1 video (id 118) from the colonoscopy sequences. For each video, only a certain number of frames is used. The number of frames used from each video changes depending on the video and the use of tools in this video. It also changes depending on the task, segmentation or classification, since the segmentation task only uses frames with tools and the classification task also includes frames without. We therefore create two sub-datasets : **HCULB-tools** and **HCULB-full**. HCULB-tools contains only frames with tools, whereas HCULB-full contains the same frames with tools from HCULB-tools and added frames without tools.

---

3. http://www.hcuz.es/

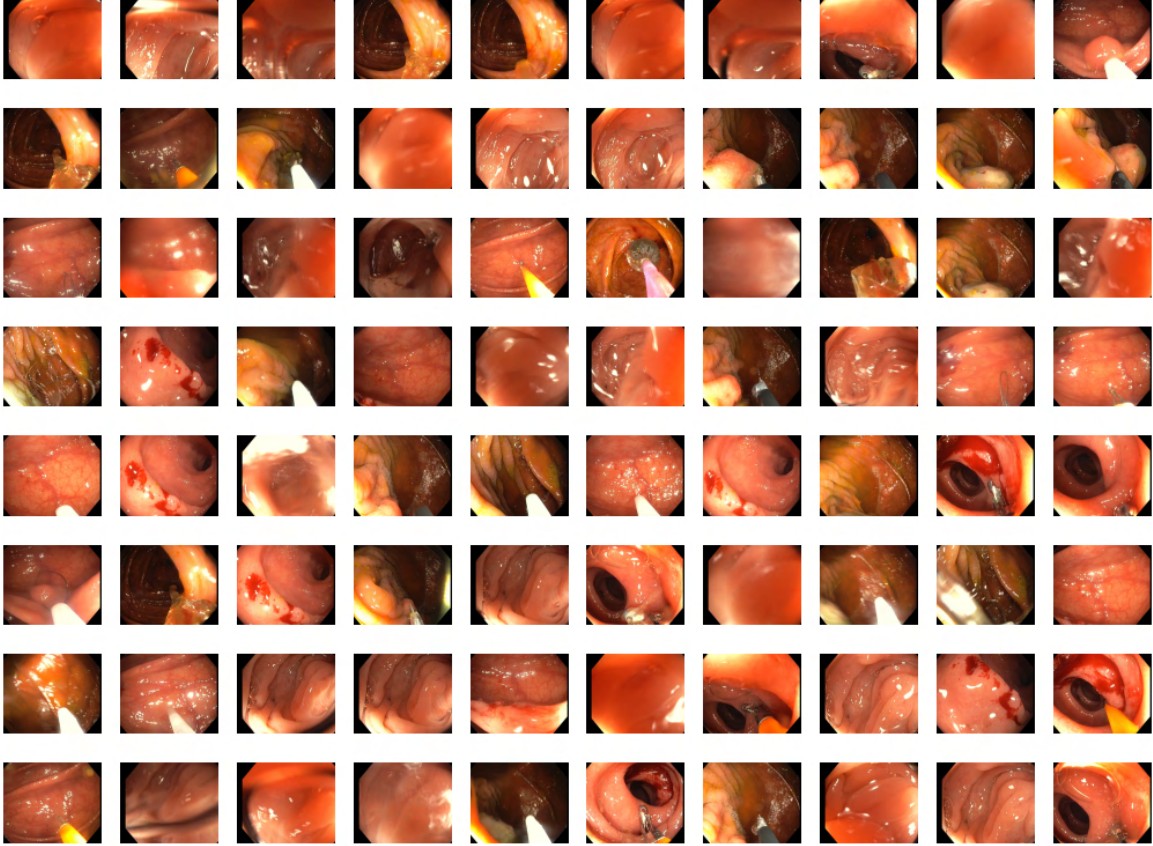

Figure 4: Examples of frames from the HCULB-full dataset

Figure 4 presents examples of frames found in the HCULB-full dataset. The frames in which tools are present can also be found in the HCULB-tools dataset.

Regarding the HCULB-tools dataset, each training sequence contains only one or two types of tool, and the type of tool seen in each sequence only appears in that sequence. The sequence reserved for evaluation is the one that contains all the different types of tool seen during training. The tools seen in the sequences are the most common tools used during procedures from HCULB. Table 4 presents the total number of frames in each set (train and test) depending on the sub-dataset. Table 5 presents the number of frames used from each video depending on the sub-dataset. The frames with tools used in HCULB-tools are the same ones used in HCULB-full.

Table 4: Total number of frames in each set depending on the sub-dataset

| Set | Task | Frames |
|---|---|---|
| Train | HCULB-tools | 825 |
| Train | HCULB-full | 1650 |
| Test | HCULB-tools | 254 |
| Test | HCULB-full | 508 |

Table 5: Number of frames used for each video and each sub-dataset

| Video id | Task | Frames |
|---|---|---|
| 39 | HCULB-tools | 161 |
| 39 | HCULB-full | 322 |
| 118 | HCULB-tools | 254 |
| 118 | HCULB-full | 508 |
| 181 | HCULB-tools | 435 |
| 181 | HCULB-full | 870 |
| 206 | HCULB-tools | 229 |
| 206 | HCULB-full | 458 |

## A.3. Effect of fine-tuning

Table 6 sums up the values of mIoU measured for the HCULB-tools test set using all models before and after fine-tuning on the HCULB-tools train set. As expected, fine-tuning allows for better adaptation to the target domain.

Table 6: Binary segmentation results (mIoU) for models only pre-trained on EndoVis17 and models pre-trained on EndoVis17 and fine-tuned on HCULB-tools

| Model | mIoU before fine-tuning | mIoU after fine-tuning |
|---|---|---|
| U-Net | 12.31 | 55.63 |
| LinkNet | 54.12 | 60.54 |
| MF-TAPNet | 27.99 | 66.87 |
| MiniNet | 37.66 | 66.65 |

## A.4. MiniNet architecture

MiniNet is a segmentation model designed specifically to be more efficient than most state-of-the-art models, where efficiency means lower number of parameters and lower inference time. It reaches similar to better results than state-of-the-art efficient models on public benchmarks. MiniNet follows an encoder-decoder structure formed by four blocks : down-sample, feature extractor, refinement and upsample blocks.

The efficiency of the model is reached using specific types of convolutional layers : depthwise separable convolutions and multi-dilation depthwise separable convolutions. The depthwise separable convolution splits a normal convolution into two convolutions. First, a depthwise

convolution is applied, which is a convolution applied to the input without modifying its depth or number of channels. One kernel is applied to each channel of the input. Then, a pointwise convolution is applied. It is a 1x1 convolution that combines all channels of the inputs, applied as many times as the desired number of channels of the output. The division of the classic convolution into two simpler convolutions allows for an important reduction of the number of learning parameters of the model, 88% in the case of MiniNet. The multi-dilation depthwise separable convolution is introduced in (Alonso et al., 2020). It is formed by two parallel depthwise convolutions and a pointwise convolution applied to the combined outputs of both depthwise convolutions. Each depthwise convolution has a different dilation rate. The dilation rate defines the step size between each of the values in the kernel used. The first depthwise convolution uses a dilation rate $r = 1$, meaning it is a classic depthwise convolution as describe before, with no spacing in between the values in the kernel. The other depthwise convolution uses $r \geq 1$, meaning there might be a gap defined by $r$ between each consecutive value in the kernel. The use of the multi-dilation depthwise separable convolution allows for reduction by 87% of the number of parameters in MiniNet, compared to using a standard convolution.

These two types of convolutions are combined with upsample and downsample operations to form the four blocks of the model. The downsample block is a succession of downsample operations, represented by a max-pooling layer followed by a strided convolution, combined with 3x3 depthwise separable convolutions and residual connections. The upsample block follows the same structure but with upsample opertions, defined by transposed convolutions, instead of downsample operations. The feature extractor uses a succession of multi-dilation depthwise separable convolutions, with different dilation rate, and dropout. The refinement block is formed by two downsample operations as described previously, applied to the input image, and allows for extraction of spatial and high resolutions features, meaning it extracts additional information which the feature extractor might have missed. Figure 5 shows the detailed structure of the model for an input of size 1024x512.

| Block | Name | Type | Input | Output size |
|---|---|---|---|---|
| Downsample | d1 | downsampling | image | 512x256x16 |
| | d2 | downsampling | d1 | 256x128x64 |
| | m1 | rate=1 | d2 | 256x128x64 |
| | m2 | rate=1 | m1 | 256x128x64 |
| | m3 | rate=1 | m2 | 256x128x64 |
| | m4 | rate=1 | m3 | 256x128x64 |
| | m5 | rate=1 | m4 | 256x128x64 |
| | m6 | rate=1 | m5 | 256x128x64 |
| | m7 | rate=1 | m6 | 256x128x64 |
| | m8 | rate=1 | m7 | 256x128x64 |
| | m9 | rate=1 | m8 | 256x128x64 |
| | m10 | rate=1 | m9 | 256x128x64 |
| | d3 | downsampling | m10 | 128x64x128 |
| Feature extractor | m10 | rate=1 | d3 | 128x64x128 |
| | m11 | rate=2 | m10 | 128x64x128 |
| | m12 | rate=1 | m11 | 128x64x128 |
| | m13 | rate=4 | m12 | 128x64x128 |
| | m14 | rate=1 | m13 | 128x64x128 |
| | m15 | rate=8 | m14 | 128x64x128 |
| | m16 | rate=1 | m15 | 128x64x128 |
| | m17 | rate=16 | m16 | 128x64x128 |
| | m18 | rate=1 | m17 | 128x64x128 |
| | m19 | rate=1 | m18 | 128x64x128 |
| | m20 | rate=1 | m19 | 128x64x128 |
| | m21 | rate=2 | m20 | 128x64x128 |
| | m22 | rate=1 | m21 | 128x64x128 |
| | m23 | rate=4 | m22 | 128x64x128 |
| | m24 | rate=1 | m23 | 128x64x128 |
| | m25 | rate=8 | m24 | 128x64x128 |
| Ref | d4 | downsampling | image | 512x256x16 |
| | d5 | downsampling | d4 | 256x128x64 |
| Upsample | up1 | upsampling | m25 | 256x128x64 |
| | m26 | rate=1 | up1 + d5 | 256x128x64 |
| | m27 | rate=1 | m26 | 256x128x64 |
| | m28 | rate=1 | m27 | 256x128x64 |
| | m29 | rate=1 | m28 | 256x128x64 |
| | output | upsampling | m29 | 512x256xN |

*rate*: stands for the convolutional dilation rate.

Figure 5: Detailed architecture of MiniNet for input size 1024x512, obtained from (Alonso et al., 2020)

## A.5. Additional inference time results

Inference time can also be measured for the encoder and decoder parts of the models, as well as with and without the pre-filtering step. Table 7 shows these results. The pre-filtering classification step from already extracted features takes 2ms.

Table 7: Detailed inference time results (ms) for the three best segmentation models and their encoder and decoder, with and without pre-filtering

| Model | Encoder inf. time (ms) | Decoder inf. time (ms) | Inf. time no pre-filtering (ms) | Inf. time with pre-filtering (ms)* |
|---|---|---|---|---|
| MiniNet | 15 | 11 | 26 | 28 |
| MF-TAPNet | 74 | 81 | 155 | 157 |

*: Maximum inference time, achieved if tool is detected and decoder is applied;
if not detected, output is zero matrix.

## A.6. Additional experiment : multi-class tool segmentation

Besides binary segmentation, another possible task is multi-class tool segmentation. In particular, for the EndoVis17 multi-class dataset there are 8 classes, where each class represents a different type of tool (1-7) or the background (0). In this experiment, all models were trained and evaluated with the corresponding training and evaluation sets from EndoVis17 multi-class dataset. Table 8 presents the results obtained, showing that MiniNet is better than state-of-the-art models in terms of memory and inference time but also significantly in accuracy for this case of Multi-class segmentation. Figure 6 shows a few examples of images from EndoVis17 dataset segmented using MiniNet.

(a)
(b)
(c)

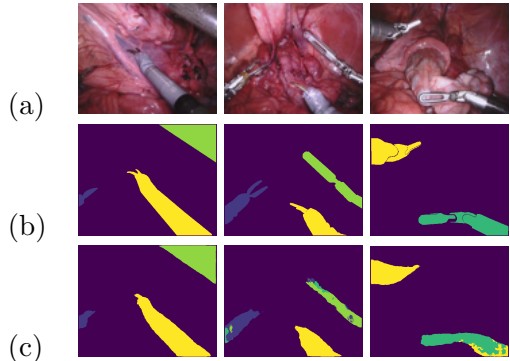

Figure 6: Examples of **Multi-class Segmentation** (8 classes) for EndoVis17 dataset : (a) Original image (b) Ground-truth manual segmentation (c) MiniNet segmentation.

Table 8: **Multi-class segmentation** (8 tool types) results (**mIoU**) on EndoVis17.

| Models | EndoVis17 | Params (M)$^\$$ | Time (ms) $^+$ |
|--------|-----------|------------------|------------------|
| U-Net | 15.80 | 7.85 | 54 |
| TernausNet | 33.78 | 36.92 | 119 |
| LinkNet | 22.47 | 21.79 | 34 |
| MF-TAPNet | 36.62 | 37.73 | 155 |
| DMNet | 61.03 | 4.38 | 183 |
| MiniNet | 62.49* | 0.52 | 26 |

*: Results for model retrained in this work as explained. Otherwise, mIoU reported by authors.

$^\$$: *Params*: Memory required by the model (M = millions of parameters).

$^+$: *Time*: Average inference time for 1 image on GPU RTX2080.

