# OpenReview forum: "Efficient tool segmentation for endoscopic videos in the wild"
_MIDL.io/2022/Conference — MIDL 2022_

### Official Review · Reviewer_UVwW · 2022-01-24

**Confidence:** 4
**Preliminary Rating:** 3
**Recommendation:** Poster

**Summary:**

The paper describes a method for tool segmentation in endoscopic images based on a DL approach. The author's former study mainly inspires the proposed model (MiniNet), and it is based on an encoder-decoder architecture for semantic segmentation. In the presented version, a new classification head is added to the model to detect images without any tools inside. The newly added head improved the performance and led to a faster inference for images without any tools inside.

**Strengths:**

 - Paper is well-written and easy to follow in all parts.
 - The proposed method shows a simple yet effective approach to enhance the segmentation and inference speed for the cases without any tools inside the frames.

**Weaknesses:**

- The proposed model has minimal technical novelty. The auxiliary classification head in a DL-based segmentation model has been proposed in former studies. The segmentation model has a very similar architecture to a conventional encoder-decoder-based model. As stated by the authors, it has a very similar architecture to the previously published MiniNet model.

**Deanonymize Review:**

no

**Detailed Comments:**

Please refer to the "Questions To Address In The Rebuttal"  and the "Weaknesses" sections.

**Final Rating After The Rebuttal:**

4: Weak Accept

**Justification Of The Final Rating:**

I would like to thank the authors for revising the manuscript. Most comments have been addressed adequately.  The technical novelty of the presented work is still very limited. However, application-wise, this work is worth to be published.

**Paper Type:**

validation/application paper

**Questions To Address In The Rebuttal:**

- As the main application contribution of the paper is related to detecting images without any tools inside, and this is only evaluated on the private HCULB dataset, it would be not easy to reproduce the results. Is the dataset going to be published along with this paper so the other researchers can use it to reproduce the results (or use the dataset as a new benchmark)?
- Please add quantitative results in Table 2 to compare the inference time (with and without pre-filtering).
- For the frames with tools, the inference time would be more as a new classification head is added to the MiniNet model. Please add quantitive results to compare  inference time for the original MinNet model and the MiniNet with a classification head (for images with tools inside)
- Further details should be added to explain the utilised MinNet model (maybe in the supplementary materials).
- The quality of figure 2 should be enhanced.

**Special Issue:**

no

---

### Official Review · Reviewer_56KF · 2022-01-24

**Confidence:** 4
**Preliminary Rating:** 2
**Recommendation:** Poster

**Summary:**

In this paper, the authors aim at tackling two common problems in endoscopic tool segmentation in real-world scenarios: Computational efficiency and being capable of dealing with frames without a tool present. Therefore, they apply a previously known efficient architecture in the field and couple it with an additional classification head to identify frames without a tool in it. By sharing the features between this classification and the segmentation task, this pipeline saves valuable computation time.
The presented approach is evaluated using three datasets, where one of them is less curated and therefore more realistic. The authors benchmark against four different state-of-the-art models without pre-filtering and against the optimal one of them with pre-filtering.

**Strengths:**

A clear advantage of this manuscript is its application-driven focus and the dedication to make an impact in realistic scenarios. The presentation of pre-existing models and datasets is profound and focussed on the important key aspects.

**Weaknesses:**

My main concern is the inconsistent storyline. For me, the methodological contribution as well as the results do not match the claims raised in the introduction. There is a huge variety of approaches to improve the computational efficiency of deep neural networks, ranging from iterative pruning to decreased resolutions of the network parameters. None of these were addressed, not even as related work. Hence, I feel that "Efficient tool segmentation" is a big claim for "applying a known architecture that works all in autonomous driving".
Similarly, the authors present the model with pre-filtering as an adapted, novel architecture, which is kind of misleading. In my opinion, it is more a pipeline with two different models, a classifier and a segmentation network. The key advantage is that both networks share the same encoder, and this increases the computational efficiency during training (classifier uses encoder from segmentation model) and inference (segmentation uses extracted features from classifier). But it is still not a new, end-to-end architecture.
Both perspectives on the problem open up many questions regarding different comparisons, and only very few of them were addressed in the manuscript. Please find detailed comments on this in the next section.

**Deanonymize Review:**

yes

**Detailed Comments:**

As mentioned above, I have several major issues to address:

1) As mentioned above, I think the MiniNet+Prefiltering architecture is not an architecture per se but two models that share the weights in their encoders. Therefore, training and Inference is more of a pipeline than a model. Hence, Fig. 2 is kind of misleading and also the wording should be adjusted.
2) Related to this point, I am missing a sufficient evaluation of this approach. Taking the more general pipeline point of view described above, the authors basically propose a wrapper framework that could be used with any encoder-decoder-based segmentation model. This is a good thing and a way bigger contribution than what the author state, but it should also be evaluated in this way. Hence, I would expect an evaluation using the other networks as well (with their respective encoder-module). Then, the accuracies as well as the processing time could be quantified for all of them to find the one with optimal balance.
3) The two challenges the authors aim to address differ between the abstract (efficiency and false positives) and the introduction (efficiency and domain adaptation). This should be consistent.
4) Related to point 3), I don't see the contribution of this work regarding domain adaptation. Even though all models are fine-tuned on the Kvasir and HCULB datasets, I see this more as an application-driven pre-training (which is quite common in deep learning) than an attempt to tackle domain adaptation. Specifically since no evaluation of this fine-tuning is performed, the effect of this transfer learning is not quantified or assessed in any way. Therefore, I have problems with stating to tackle domain adaptation in this study.
5) How was the pre-filtering in Table 2 performed on MF-TAPNet? Was the binary classifier the same as in MiniNet (using the MiniNet encoder) or was the encoder part of MF-TAPNet used for it? This should be more clear.
6) Related to point 5), I miss an explanation on the pre-filtering results presented in Table 2. How can MiniNet reach a comparable accuracy to Table 1 if the filter only catches 72% of frames that don't show a tool? Since the segmentation itself does not change, I would expect a greater gap between the accuracy on tool-frames only (Table 1) and the one on filtered images (Table 2). Specifically since this gap is larger for the MF-TAPNet, I am missing some discussion here.

In addition, I have some minor comments:
1) On page 2, line 2, it is not clear what algorithm produced the figures. Does "this work" refer to the manuscript or the cited study?
2) In 2.2, the architecture of MiniNet it described quite superficially. Since applying MiniNet in tool segmentation is the core contribution of this work, it should be described in more detail why the model is s efficient.
3) Later on in 2.2 under "Binary frame classification module", the authors state that they also tried other approaches to avoid false-positives but do not give details of the results. Since they present the classification header as a solution to one of the two big problems in real world tool segmentation (following the author's claim), these comparisons are crucial to highlight the value of their proposed approach. Hence, the quantitative results of the other approaches should be given in the manuscript for comparison.
4) In Table 1, why are there no results for the TernausNet on the Kvasir and HCULB data set?
5) In Fig. 5, I assume that the results from MiniNet are achieved with a "vanilla" MiniNet without pre-filtering? It would be interesting to see the comparison with and without filters.

**Final Rating After The Rebuttal:**

4: Weak Accept

**Justification Of The Final Rating:**

The authors addressed all my concerns adequately and thoroughly revised their manuscript, The wording of the paper was greatly improved and the revision increased the quality significantly.
While the methodological novelty remains moderate, the work presents a relevant contribution from an applied point of view and I am sure that the proposed concepts and results will be of great interest for experts from the field of surgical robotics.
Hence, I am happy to vote for accepting the manuscript and congratulate the authors on their nice contribution to bring deep learning in surgical robotics to the real world.

**Paper Type:**

validation/application paper

**Questions To Address In The Rebuttal:**

As mentioned above, I really appreciate the value of bringing tool segmentation into the real world and I think that the results are of interest to the application-driven community. However, from a scientific point of view, I think that the manuscript itself describes the methods in a slightly skewed way and presents results that do not back up the claims. Therefore, I cannot vote for accepting this manuscript in its current state.
However, I want to encourage the authors to make good use of the rebuttal time! Many weaknesses I listed above can be solved by rephrasing and rearranging parts of the work, and I feel that at least some of the results I would wish to see here are already there and were just not listed by the authors. Hence, a thorough revision could lead to a substantially improved manuscript, and I am looking forward to it!

**Special Issue:**

no

---

### Official Review · Reviewer_n8QZ · 2022-01-25

**Confidence:** 4
**Preliminary Rating:** 2
**Recommendation:** Poster

**Summary:**

This work presents a neural network architecture for robotic tools segmentation from endoscopy images that tackles the problem of the false positives.
The architecture here presented is inspired by MiniNet (Alonso, ECCV 2020) and is characterized by a low number of parameters and a short inference time.
A classification head is added to the architecture, to determine the presence or absence of the tools is inserted and so determine the presence or absence of the tool in the image.

**Strengths:**

- The presented architecture achieves performance comparable to other architectures present in the state of the art with a very low number of parameters and a shorter inference time
- The addition of a classification head allows a significant reduction in the number of false positives.
- The comparison of the presented architecture with the other selected ones has been performed on multiple datasets

**Weaknesses:**

- No innovation was introduced by the presented work, the author merely applied an existing architecture (MiniNet) on medical dataset.
- Despite the significant reduction in the number of parameters, the inference time compared to LinkNet is not significantly lower (about 9ms less) considering the usual framerate of 25/30fps
- The addition of the classification head shows a reduction in false positives, but does not introduce a particular methodological novelty. Moreover, for this variation the results have been compared only with one architecture among those presented.
- Some details of the training protocol are incomplete.
- HCULB dataset is not public

**Deanonymize Review:**

no

**Final Rating After The Rebuttal:**

3: Borderline

**Justification Of The Final Rating:**

The authors addressed all my concerns and revised their manuscript. The writing quality of the paper was greatly improved and the revision increased the overall quality. Some minor improvements to the layout are still required.
The work presents an interesting contribution since is applied in the wild and for the focus on the efficiency, however the methodological novelty is still very limited.

**Paper Type:**

validation/application paper

**Questions To Address In The Rebuttal:**

- It would be helpful to have more information about the training protocol: What are the image sizes used to train the networks used in this work? How long does it take to train the architecture used compared to the others? The networks are trained from initial weights (e.g. ImageNet) or entirely from scratch?
- It is worthy of performing a comparison with architectures with a reduced number of parameters such as ERFNet/ESPNet
- Introduce the environmental benefit of using more efficient architectures. I recommend exploring this topic further from Carbon Emissions and Large Neural Network Training (Patterson et al.)
- The HCULB dataset is not public. It would be interesting to present an analysis of the dataset to understand the degree of complexity of the dataset. It would also be helpful to have more visual examples of the dataset.
- Figure 1 is too small
- Figure 2 is particularly grainy. In addition, provide more detail on the network structure in the figure.
- I would compare the inclusion of the classification head on all architectures

**Special Issue:**

no

---

### Meta-Review · Area_Chair_VMmQ · 2022-02-18

**Recommendation:** Accept (Poster)
**Confidence:** 5

**Metareview:**

The work develops a method for tool segmentation in endoscopy videos, with 2 primary goals: Make the method more efficient towards achieving real-time inference; reduce false positives, in particular in frames where tools aren’t visible.

Strengths
* (as nicely put by Reviewer 56KF) “its application-driven focus and the dedication to make an impact in realistic scenarios.”
* Results that support the claims (faster, lower false positives)
* Evaluation on real-world data that may be more challenging than common benchmarks in this field.

The main concerns raised by reviewers were:
* Not much methodological novelty to the field of Machine Learning, but rather an effective application of existing methods (MiniNet with an additional classification head).
* Relatively limited evaluation / comparisons with other models.
* Issues with clarity of methodology and evaluation.
* Some issues with presentation.
* Evaluation on non-public dataset that may hinder reproducibility / future comparisons. (Though the authors state in their answers that they are in the process of getting approval for publishing the data. This would be very useful to the community if it happens.)

During the rebuttal/discussion period and the corresponding updates to the paper, the authors addressed quite some of the reviewers’ concerns, especially with respect to clarity of methodology, evaluation, contribution, and presentation. This is reflected with the updated scores and comments by the reviewers, who seem to agree that in its updated version, the paper is of content and quality acceptable for publication.

---

### Decision · Program_Chairs · 2022-02-28

Accept